# Combining Genetic and Transcriptomic Approaches to Identify Transporter-Coding Genes as Likely Responsible for a Repeatable Salt Tolerance QTL in Citrus

**DOI:** 10.3390/ijms242115759

**Published:** 2023-10-30

**Authors:** Maria J. Asins, Amanda Bullones, Veronica Raga, Maria R. Romero-Aranda, Jesus Espinosa, Juan C. Triviño, Guillermo P. Bernet, Jose A. Traverso, Emilio A. Carbonell, M. Gonzalo Claros, Andres Belver

**Affiliations:** 1Instituto Valenciano de Investigaciones Agrarias (IVIA), 46113 Valencia, Spain; raga_ver@gva.es (V.R.);; 2Department of Molecular Biology and Biochemistry, Universidad de Málaga, 29010 Malaga, Spain; amandabullones@uma.es (A.B.); claros@uma.es (M.G.C.); 3Integrative Biology for Plant Stress Group, La Mayora Institute of Subtropical and Mediterranean Horticulture, IHSM-CSIC-UMA, 29750 Malaga, Spain; mr.romero@csic.es; 4Department of Stress, Development and Signaling of Plants, Estación Experimental del Zaidín, Consejo Superior de Investigaciones Científicas (EEZ CSIC), C/Prof. Albareda 1, 18008 Granada, Spain; jesus.espinosa@eez.csic.es (J.E.); andres.belver@eez.csic.es (A.B.); 5Sistemas Genómicos S.L., Ronda de Guglielmo Marconi, 6, 46980 Paterna, Spain; jc.trivino@sistemasgenomicos.com (J.C.T.); guillermo.bernet@sistemasgenomicos.com (G.P.B.); 6Department of Cellular Biology, Faculty of Sciences, Universidad de Granada, 18071 Granada, Spain; traverso@ugr.es; 7CIBER de Enfermedades Raras (CIBERER) U741, 29071 Málaga, Spain; 8Institute of Biomedical Research in Málaga (IBIMA), IBIMA-RARE, 29010 Málaga, Spain

**Keywords:** QTL analysis, *Citrus reshni*, *Poncirus trifoliata*, rootstock breeding, yield, Cl^−^ homeostasis, root growth, plasticity

## Abstract

The excessive accumulation of chloride (Cl^−^) in leaves due to salinity is frequently related to decreased yield in citrus. Two salt tolerance experiments to detect quantitative trait loci (QTLs) for leaf concentrations of Cl^−^, Na^+^, and other traits using the same reference progeny derived from the salt-tolerant Cleopatra mandarin (*Citrus reshni*) and the disease-resistant donor *Poncirus trifoliata* were performed with the aim to identify repeatable QTLs that regulate leaf Cl^−^ (and/or Na^+^) exclusion across independent experiments in citrus, as well as potential candidate genes involved. A repeatable QTL controlling leaf Cl^−^ was detected in chromosome 6 (*LCl-6*), where 23 potential candidate genes coding for transporters were identified using the *C. clementina* genome as reference. Transcriptomic analysis revealed two important candidate genes coding for a member of the nitrate transporter 1/peptide transporter family (NPF5.9) and a major facilitator superfamily (MFS) protein. Cell wall biosynthesis- and secondary metabolism-related processes appeared to play a significant role in differential gene expression in *LCl-6*. Six likely gene candidates were mapped in *LCl-6,* showing conserved synteny in *C. reshni.* In conclusion, markers to select beneficial Cleopatra mandarin alleles of likely candidate genes in *LCl-6* to improve salt tolerance in citrus rootstock breeding programs are provided.

## 1. Introduction

Salinity in soil and irrigation water constitutes a serious global threat to food security, affecting 2.1% of dry agricultural land and up to 19.5% of irrigated land, which accounts for one-third of world food production [1]. Losses in agricultural production globally due to salinity are estimated to be USD 27 billion per year [2]. Citrus is ranked among the most salt-sensitive tree crops [3]. Tree growth and fruit yield of citrus species are impaired at a soil salinity of approximately 2 dS/m soil saturation, without any concomitant expression of leaf symptoms [4,5]. Additionally, salinity predisposes the citrus tree root to attacks by *Phytophthora* [6], nematodes [7], and bacterial pathogens [8]. As citrus varieties of sweet oranges, mandarins, grapefruits, pummelos, and lemons are always propagated by bud-grafting onto a seedling rootstock in order to obtain a uniform orchard with tolerance to soil pathogens and well-adapted to local edaphoclimatic conditions, salt tolerance is often a target trait in citrus rootstock breeding programs [9]. In addition, the root (contributed by the rootstock) is the first plant organ to come into contact with salts and plays a major role in water and nutrient uptake.

However, citrus breeding for salt tolerance is not an easy task due to the following factors: (1) salt tolerance is a complex, quantitative trait that depends on the level and duration of salinity, its definition (vegetative growth, fruit yield), and the progeny (parents) under study; and (2) the long juvenility period of citrus (a mode value of 8 years; [10]) and segregation for nucellar embryony in citrus rootstock progenies [10]. Fortunately, the rich citrus germplasm has great halotolerant genetic potential to improve citrus rootstocks [11,12,13,14,15,16,17].

Citrus species used as rootstocks were originally classified into three groups [18]: *Citrus reshni* (Cleopatra mandarin), with a good level of salt tolerance; *Citrus volkameriana* (Volkamer lemon) and *Citrus aurantium* (sour orange), with a medium level of salt tolerance; and *Poncirus trifoliata* (trifoliate orange), with low salt tolerance levels. The literature on the ranking of species since that time has been abundant, and the differences observed depend on the specific species and the set of accessions per species under study. However, Cleopatra mandarin has always been ranked at the top of these classifications [11,13,14], which is useful for breeding improved rootstocks [11,17]. The identification of beneficial gene alleles in the halotolerant germplasm using QTL analysis of salt tolerance would facilitate their introduction into new citrus rootstocks. This would enable marker-assisted selection, make breeding programs more efficient—particularly in relation to fruit trees with long juvenility—and make the genetic modification of this complex trait feasible.

Under saline conditions, in several fruit crops, such as citrus, grapevine, avocado and persimmon, the excessive accumulation of Cl^−^ in leaves (but not of Na^+^) has been found to be related to decreased transpiration, photosynthesis, yield and quality of crops, and eventually the death of the plant [19,20,21,22,23,24]. All the data indicate that the accumulation of Cl^−^ ions in shoots is more often associated with a reduction in the growth and yield of citrus under salinity conditions than with the accumulation of Na^+^ in shoots. However, the genetic regulation of the movement of Cl^−^ from the root to the shoot as compared to that of Na^+^ has been rarely studied [25]. Cl^−^, which is an essential micronutrient for plants and acts as a cofactor in photosynthesis and in numerous enzymatic activities. However, given its mM accumulation levels in plants, Cl^−^ is also a beneficial macronutrient and is involved in osmotic functions, affecting key processes such as stomatal movement, charge balance, and cell expansion [26,27]. In fact, low concentrations (1–5 mM) of Cl^−^ in irrigation water increase water (WUE), nitrogen (NUE), and CO_2_ use efficiency in well-watered tobacco plants and improve their ability to withstand drought [26,27,28]. However, under saline conditions, a reduction in Cl^−^ concentrations in the xylem is the key step in reducing Cl^−^ toxicity in the aerial part, which can be achieved by limiting the entry of Cl^−^ into the xylem and/or by enhancing its removal [23,25]. Genomic-wide transcriptomic information on putative Cl^−^ transporters involved in citrus salt tolerance is limited [20,29,30], particularly regarding the root [29,30]. The experimental design of these transcriptomic studies based on just one genotype [29,30] or two genotypes from different species [20] prevents distinguishing heritable tolerance responses from global responses or species-specific responses to salinity.

The detection of genomic regions containing QTLs controlling net Cl^−^ entry into the xylem could play a useful role in the search for candidate genes coding for the transporter(s) involved. Early attempts at QTL analysis of this trait have been reported by Tozlu et al. [31] and Raga et al. [17]. However, genetic studies of citrus are usually based on a small number of genotypes, which greatly affects the power of detection of QTLs, even after increasing the number of replicates per genotype by using nucellar seedlings. This makes the repeatability of QTLs across experiments difficult to observe. However, the repeatability of a QTL is essential to ensure the success of marker-assisted selection. This study aims: (1) to find a repeatable QTL controlling leaf Cl^−^ and/or Na^+^ exclusion through independent experiments using a reference progeny (R×Pr) derived from two well-known citrus rootstocks, Cleopatra mandarin (R) and trifoliate orange (Pr), differing in salt tolerance; (2) to identify potential candidate genes coding for transporters belonging to large gene families underlying the QTL; and (3) to study differential gene expression at the root of two contrasting salt-tolerant members of the progeny for the genes in the confirmed QTL in order to further rank the candidate genes. In this original approach, we describe the integration of QTL and RNA-seq analyses to identify transporter coding genes as likely involved in the leaf chloride exclusion mechanism of the salt-tolerant Cleopatra mandarin. The markers provided will be useful to select the beneficial Cleopatra alleles to improve salt tolerance in citrus rootstock breeding programs and to set up core collections of *Citrus* and *Poncirus* accessions where new salt-tolerant materials might be more easily found.

## 2. Results

This study covers the results obtained from two long-lasting salt tolerance experiments using the same mapping population, named R×Pr. In one experiment, a citrus commercial variety was grafted onto the nucellar seedlings from apomictic R×Pr hybrids (grafted population, GP experiment, Appendix A). In the other experiment, new nucellar non-grafted seedlings from apomictic R×Pr hybrids were used (NG experiment, Appendix A).

### 2.1. Salt Tolerance in the GP Experiment

Under salinity conditions, the estimated heritabilities for some traits, such as leaf [Cl^−^], fruit soluble solids content, and total fruit yield (Cl_L_S, SSC_S, and TFW_S, respectively), were substantial (higher than 0.3) in this experiment (Appendix A). Salt tolerance, measured as TFW_S, was highly significantly correlated with Cl_L_S (r = 0.59; Appendix A). In addition, TFW_S was positively correlated with TChl_S and SSC1_S and negatively correlated with K_L_S and Na_L_S. The distributions of relevant traits evaluated in this experiment are shown in Appendix A, in which the means of the parents (Cleopatra, Flying Dragon) and the reference rootstock Carrizo are also included.

A few significant QTLs are shown in Table 1, most of which correspond to Cleopatra segregation. A clustering of QTLs for fruit yield traits, leaf concentrations of Cl, Fe, and K under salinity conditions, and soluble solid content (SSC) of fruit juice, also under salinity conditions, was observed in linkage group 4c (*C. clementina* scaffold 6) near marker CR23,750 (in Ciclev10013177m.g), where a major QTL (percentage variance explained (PVE) = 52%) controlling Cl_L_S was detected (Appendix A). The directions of correlations mostly fitted genotypic means at CR23,750, with Cleopatra allele **a** (in genotypes **ac** and **ad**) being associated with low Cl_L_S and high TFW, NFp, and Fe_L_S, while allele **b** was associated with high Cl_L_S and low fruit yield (Appendix A). Therefore, salt tolerance conferred by the R×Pr rootstock on the satsuma variety in terms of fruit yield is genetically related to the leaf Cl^−^ concentration, as QTLs for both traits were found to be located in the same genomic region around marker CR23,750 in this experiment.

### 2.2. Salt Tolerance in the Non-Grafted Population (NG Experiment)

A very wide distribution of leaf Cl concentrations under salinity conditions was observed as compared to that for the root (Appendix A). In the plant, under control conditions, Cl^−^ concentrations are generally higher in the root than in the leaf with respect to most of the hybrids; however, under salinity conditions, around 50% of the population presented higher Cl^−^ levels in the leaf than in the root (Appendix A), meaning an excess.

Cl^−^ was transported to the leaves, where it then accumulated. Eight hybrids, including hybrid 107, showed no significant differences in the distribution of plant Cl^−^ between salinity levels and showed similar leaf Cl^−^ levels to those of the salt-tolerant parent, Cleopatra (Figure 1C). These hybrids are considered salt tolerant. Moreover, similar differences in leaf Cl^−^ levels between salt-sensitive and tolerant genotypes (trifoliate orange versus Cleopatra, and 90 versus 107) under salinity (Figure 1A) were also found in the previous (GP) experiment (Table 2) where leaves belong to a grafted mandarin.

Results from the mixed model analyses and estimated heritabilities are shown in Appendix A. As expected, G×E interactions were highly significant (*p* < 0.0001) for Cl^−^-related traits, thus indicating important differences between the behaviors of hybrids when salinity levels change. Differences in phenotypic plasticity with respect to Cl^−^ traits can be easily observed as differences in the slopes of their reaction norms, particularly for leaf [Cl^−^] and the leaf-to-root distribution of Cl^−^ concentrations, where the salt-tolerant hybrids showed almost no change (Appendix A). Thus, salt tolerance is associated with stability (robustness) in leaf and leaf-to-root Cl^−^ concentrations.

Under salinity conditions, leaf [Cl^−^] is positively related to leaf [Na^+^] and [Ca^2+^] and negatively related to N_L_S (Appendix A). In general, root mass, total root dry weight (TRDW), and fine root dry weight (FRDW) are positively related to leaf [Cl^−^], meaning that the larger the root, the more Cl^−^ is accumulated in the leaf. It is noteworthy that, under salinity conditions, TRDW and FRDW are additionally related (negatively) to root [Cl^−^]; this means that the larger the root, the more Cl^−^ is accumulated in the leaf and the less Cl^−^ is accumulated in the root. FRDW increased under salinity conditions, an increase which was particularly high in some hybrids (Appendix A), making the G×E interaction significant for this trait (Appendix A).

A list of significant QTLs is presented in Table 3. Most QTLs corresponded to the segregation of Cleopatra alleles, with, again, a clustering of QTLs being observed in linkage group 4c (Appendix A). Again, the QTL for leaf [Cl^−^], henceforth named *LCl-6,* which had a particularly strong effect (39.4%) under salinity conditions, was located in the genomic region between markers 15R,750 at Ciclev10011720m and CR28,270 at Ciclev10011175m in *C. clementina* scaffold 6. The QTLs influencing root dry mass (TDRW_C and FRDW_C) and leaf Ca, C, and Fe (Ca_L_S, C_L_S, and Fe_L_S) were also present in this genomic region. Salt tolerance in terms of leaf Cl content was again associated with the Cleopatra allele **a** at marker CR23,750 (genotypes **ac** and **ad** in Appendix A).

### 2.3. Genomic and Transcriptomic Analysis of Genes in the Salt Tolerance QTL LCl-6

A list of candidate genes coding for transporters that might be involved in leaf Cl^−^ exclusion/accumulation was drawn up for *LCl-6*, with *C. clementina* as the reference genome (Table 4). It is noteworthy that the region between markers CR23,750 and 15R,750 included seven genes from the nitrate transporter 1/peptide transporter (NPF) family: NPF2.3 (Ciclev10013496m), NPF5.12 (Ciclev10011341m), NPF8.1 (Ciclev10011381m), NPF8.2 (Ciclev10013821m), NPF5.9 (Ciclev10013337m), NPF5.8 (Ciclev10013636m), and NPF5.10 (Ciclev10013488m). The same number of genes from this NPF protein family was found on the downloaded list of candidate genes, with *Poncirus trifoliata* as the reference genome (Appendix A).

The root transcriptomics of two full-sibs from the R×Pr population, which differed considerably both in terms of salt tolerance (Figure 1 and Appendix A) and of the Cleopatra allele at *LCl-6* (numbered 107 and 90, salinity-tolerant and -sensitive, respectively), were comparatively studied by bioinformatically analyzing the RNA sequencing in the NG experiment after 383 days of salt treatment (Appendix A). Taking into account the whole genome of *C. clementina* as reference, around 80% of genes were expressed at the root under our experimental conditions; in addition, the number of differentially expressed genes (DEGs) depended on the method used for their assessment (iDEP versus RSeqFlow with the use of an eBayes or treat procedure) and the comparison under consideration as shown in Table 5. Most DEGs corresponded to the comparison between hybrids and their responses to salinity ((107_15 + 107_0) vs. (90_15 + 90_0)), followed by the comparison between hybrids under control conditions. When considering only the genes included in this QTL (452 or 465, depending on the reference genome), 322 (71%) or 303 (65%) genes were expressed at the root. From the 322 QTL-expressed genes (*C. clementina* as reference and RSeqFlow together with Treat as the procedure used), the comparison between 107_15 and 90_15 resulted in eight DEGs, the comparison between 107_0 and 90_0 resulted in 21 DEGs, while the comparison between (107_15 + 107_0) and (90_15 + 90_0) resulted in 92 DEGs. It is worth noting that no differential transcriptomic root response to salinity was hybrid-independent ((107_15 − 90_15) vs. (107_0 − 90_0)) at either the whole genome or *LCl-6* QTL level.

A total of 78 genes (68 annotated genes) in the QTL were involved in these DEGs as a whole. All DEGs and the corresponding genes in the salt tolerance QTL are listed in Appendix A. Thus, from our initial list of 23 functional candidate genes in Table 4, 11 became relevant when their differential mRNA expression at the root was tested; of these 11 genes, 4 are NPF-coding genes: Ciclev10013337m (NPF5.9), Ciclev10011381m (NPF8.1), Ciclev10013821m (NPF8.2), and Ciclev10011341m (NPF5.12). It is also worth noting that the only candidate gene whose expression in the tolerant hybrid decreased was Ciclev10011341m (NPF5.12).

Considering that *P. trifoliata* is the reference genome, some candidate genes in the QTL also showed differential mRNA expression (Appendix A). Most *Citrus* and *Poncirus* DEGs correspond to the same candidate genes except those coding for PIP2.1. Thus, gene Ptrif.0006s0418.2 (ortholog of Ciclev10012379m) is differentially expressed only when *Poncirus* is considered to be the reference genome, and gene Ciclev10012375m is differentially expressed when *C. clementina* is the reference genome. The *C. clementina* ortholog of gene Ptrif.0006s0399 (coding for a reverse transcriptase), which corresponds to a DEG in two comparisons, was absent; this suggests that differences in *Poncirus*-specific retrotransposon activity exist between the hybrids in response to salinity at the root.

Once gene alleles a/b from Cleopatra and c/d from Flying Dragon could be assigned by whole genome sequencing of trees 107, 90, and 22-7, whose genotypes at CR23,750 were ac, bc, and bb, respectively, it was possible to infer the translated sequence of each allele of the differentially expressed candidate genes. The alignments of translated sequences for the alleles of Ciclev10013337m/Ptrif.0006s0823.1 (NPF5.9), Ciclev10012375m/Ptrif.0006s0419.1, coding for plasma membrane aquaporin PIP2.1, and Ciclev10011745m/Ptrif.0006s0610, coding for a major facilitator superfamily (MFS) protein, are shown in Appendix A, respectively.

The mapping of clean reads from genotypes 107 and 90 for relevant candidate genes NPF5.9, MFS, NPF5.12, and PIP2.1 is shown in Appendix A. No read from salt-sensitive genotype 90 for the MSF candidate gene was obtained (Appendix A). The gene expression of Ciclev10013337m (NPF5.9) in salt-sensitive genotype 90 is virtually zero (Appendix A), and additionally, its corresponding protein NPF5.9 is truncated (Appendix A).

The agglomerative hierarchical clustering of DEGs in *LCl-6* based on their expression patterns, represented as the normalized mean of counts per million reads mapped, resulted in three clusters (Appendix A). Genes in cluster AHC-1 exhibited a slight increase in expression under salt stress conditions in genotype 107, while decreasing in genotype 90 under similar conditions. The co-expression network of DEGs in AHC-1 is shown in Figure 2. Thus, one can clearly observe several processes usually related to plant development and environmental stress, such as cell wall biosynthesis mechanisms, specifically glucuronoxylan and xylan biosynthesis, as well as lignin metabolism, in addition to secondary metabolism (alkaloid/phytoalexin biosynthesis).

### 2.4. Incorporating LCl-6 Candidate Genes into the Linkage Maps to Improve the Mapping Resolution of QTL Analysis

The analysis of DEGs was used as a tool to rank the different functional candidate genes within *LCl-6*. Thus, the whole reference mapping population R×Pr was genotyped for the most relevant candidates, NPF5.9, MFS, NPF8.1, PIP2.1, and ABCG. It is worth noting that Flying Dragon and Rich were genotypically different regarding candidate gene NPF5.9, with just one allele in common. All the candidate genes mapped in linkage group 4c were observed to maintain *C. clementina* synteny (Appendix A). All these candidate genes, except NPF8.1, which only segregated for the *Poncirus* alleles, were mapped in *C. reshnii* linkage group R4c (Appendix A). QTL analyses of chloride-related traits evaluated in GP and NG experiments, and that previously reported [17], showed that the closest gene to the LOD peak was usually MFS by using both the cross-pollination genotyping configuration and the double haploid configuration for the female (Cleopatra) segregation (Figure 3A,B, respectively). In the case of the non-grafted experiment, leaf [Cl^−^], and the difference between root and leaf chloride concentrations relative to that of the root under salinity, (ClR-ClL/ClR)_S, LOD peaks were flatter and fell between MFS and ABCG. It is worth noting that LOD scores at QTL peaks of these Cl-related traits increased when the candidate genes were incorporated into the linkage map. When Kruskal–Wallis statistic K was used to study marker–trait associations, similar results were obtained (Figure 3C). In the case of (ClR-ClL/ClR)_S, the K values for MFS, ABCG, and PIP2.1 were similar (16.486, 16.327, and 16.707, respectively).

The resolution of QTL analysis of fruit yield (TFW, NFp), root mass (TRDW, FRDW) under control conditions, as well as leaf Fe concentration under salinity conditions (GP_L_Fe_S), was improved by incorporating the candidate genes into the linkage map (Appendix A). Thus, the LOD profile for GP_L_Fe_S reached a maximum exactly at NPF5.9, where TFW_S and NFp_S also displayed one of two peaks. Similarly, root growth under control conditions clearly peaked around ABCG. On the other hand, other QTLs reported in linkage group 4c, such as L_Ca_S and L_K_S in the GP experiment (Table 1), became non-significant.

From a practical perspective, markers at MFS and NPF5.9 (Figure 4) constitute new tools to help select beneficial Cleopatra alleles in rootstock breeding programs that use Cleopatra progenies to improve the salt tolerance of citrus plants.

## 3. Discussion

### 3.1. The Salt Tolerance QTL LCl-6 Has Been Consistently Detected across Three Experiments

There have been two previously reported QTL analyses of salt tolerance in citrus plants [17,31]. The latter study used the same R×Pr reference population of hybrids as the GP and NG experiments in this study. It is worth noting that [17] also detected a QTL for leaf [Cl^−^] in the same region as *LCl-6*, around the SSR marker CR23,750, where Cleopatra allele segregation was also involved (Cl*_S in Appendix A). Thus, the salt tolerance QTL *LCl-6* has been consistently detected across three independent experiments (one grafted and two non-grafted experiments) using the same R×Pr reference population derived from the intergeneric cross between Cleopatra mandarin (R) and *Poncirus trifoliata* (Pr). Therefore, *LCl-6* can be considered a confirmed QTL [32], whose responsible genes are acting at the root, which is in line with previous reciprocal grafting studies of citrus plants [33].

Although other QTLs for traits correlated with leaf [Cl^−^] (Appendix A) were also detected in the same genomic region as *LCl-6,* their detection generally depended on the experiment carried out (Appendix A). For example, a QTL for leaf K^+^ concentration under salinity conditions (K_L_S; Appendix A) was detected in the GP experiment and was also reported by [17]; QTLs for Fe_L_S and Ca_L_S were detected in both the GP (Appendix A) and NG (Appendix A) experiments. In addition to Cl-related traits, the *LCL-6* region contained QTLs for mandarin fruit yield (TFW, NFp) in the GP experiment (Appendix A) and for root growth (TRDW_C and FRDW_C) in the NG experiment (Appendix A). Despite the significant correlation between Cl_L_S and N_L_S (r = −0.5, Appendix A), suggesting a certain degree of competition between Cl^−^ and NO_3_^−^ under excess soil Cl^−^ conditions, no QTL for N_L_S was detected at or close to *LCl-6*.

Regarding other QTL analyses of stress tolerance traits, *LCl-6* overlaps with *CD/FS-2016-t6*, two QTLs for Huanglongbing tolerance, in terms of canopy damage and foliar symptoms, mapped to the *Poncirus trifoliata* genome [34]. In addition, these tolerance QTLs included candidate gene Ptrif0006s0773.1, coding for a laccase 7-related protein and responsive to infection by *Candidatus* Liberibacter, the Huanglongbing pathogen [35], which was also expressed in the root (ESR42493) in *LCl-6* in our salinity experiment. The overlapping of the salt tolerance QTL *LCl_6* and these Huanglongbing tolerance QTLs genetically supports the relationship between salinity and bacterial pathogen effects on citrus plants [8].

Therefore, the genomic region where *LCl-6* is located appears to contain genes that control relevant agronomic traits, ranging from root development to leaf nutrient content and fruit yield-related traits mediated by the rootstock, which merit further molecular research.

### 3.2. Salt Tolerance Mechanism(s) behind LCL-6

Both leaf Na^+^ and Cl^−^ concentrations were studied in the two salt tolerance experiments described here. Under salinity conditions, both parameters were positively correlated and negatively related to fruit yield in the GP experiment. Only a leaf Cl^−^ QTL (*LCl-6*) was consistently detected under salinity conditions in the same genomic region in independent experiments. Moreover, this region also contained a fruit yield QTL that genetically supports the correlation between leaf Cl^−^ and fruit yield in the GP experiment under salinity conditions. These results in the R×Pr population are in line with those from numerous scientific studies, which conclude that Cl^−^ toxicity is more important than Na^+^ toxicity in citrus plants and other woody species under salinity conditions [12,36,37,38,39]. The differences in Cl^−^ tolerance exhibited by plants are usually related to the ability to restrict Cl^−^ transport. This Cl^−^ transport in long-term stressed plants involves two main steps: (1) a net influx (from the influx/efflux balance) of Cl^−^ from the soil to the root and (2) a net loading (from the loading/unloading balance) of Cl^−^ onto the xylem to follow the transpiration stream up to the leaves. Regarding the comparison of the behavior of the R×Pr population with respect to the distribution of Cl^−^ between leaf and root (Figure 1C and Appendix A), although salinity increases Cl^−^ in both organs, this increment is proportionally smaller in the leaves of salt tolerance hybrids like 107 (Figure 1C). Thus, while hybrids 107 and 90 have similar root Cl^−^ levels (Figure 1B), they greatly differ with respect to leaf Cl^−^ concentrations under salinity conditions (Figure 1A), resulting in practically no change in Cl^−^ organ distribution between treatments in salt-tolerant hybrids (Figure 1C and Appendix A). Salt-sensitive hybrid 90 accumulates Cl^−^ in the leaves, while salt-tolerant hybrid 107 shows higher Cl^−^ levels in the root than in the leaves under salinity conditions, and both hybrids show similar root Cl^−^ levels. It, therefore, seems reasonable to hypothesize that the main mechanism involved here to prevent leaf Cl^−^ accumulation takes place in the first step (net influx of Cl^−^ from soil to the root). A greater influx of Cl^−^ into hybrid 90 would facilitate higher leaf Cl^−^ levels in 90, and a greater efflux of Cl^−^ out of hybrid 107 (such as reported in poplar species [38]) would explain the salt tolerance behavior of hybrid 107. However, taking into account the differences in root growth between hybrids 107 and 90 under salinity conditions (Appendix A), a simpler hypothesis, presented below, could explain the higher leaf Cl^−^ levels in hybrid 90 than in 107, as citrus leaf Cl^−^ accumulation has been reported to be linked to water use [40].

The root, which is the first plant organ to sense salinity in the soil, plays an important role in water uptake; therefore, its growth response to salinity has to be included in our hypothesis regarding leaf Cl^−^ exclusion. In general, although salinity reduces root mass [41], root growth responses to salinity reported in citrus-related literature are contradictory [29,42,43,44]; this is in line with our results regarding G×E interactions (Appendix A) and reaction norms for fine root mass (Appendix A). In our experiment, the root mass of some R×Pr hybrids, particularly that of salt-sensitive hybrid 90, increased considerably under salinity conditions (Appendix A). Since root dry mass (TRDW and FRDW) and leaf [Cl^−^] (Cl_L) are positively correlated (Appendix A), it is reasonable to hypothesize that, under salinity conditions, the larger root of hybrid 90 could simply facilitate a higher net influx of Cl^−^ from the soil to the root, and finally to the shoot, than that for hybrid 107.

Co-expression approaches have previously been used to assign function to genes involved in root elongation and other related traits because genes functioning in the same pathway or required for the same process tend to express in a transcriptionally coordinated manner [45]. Thus, following the agglomerative hierarchical clustering (AHC) analysis of DEGs in LCl-6, the co-expression network of DEGs in AHC-1 led to a slight increase in gene expression in hybrid 107 but to a decrease in hybrid 90 under salinity conditions (Appendix A). This shows that several processes related to plant development and in response to environmental stress, such as cell wall biosynthesis mechanisms, specifically glucuronoxylan and xylan biosynthesis, as well as lignin metabolism and secondary metabolism (alkaloid/phytoalexin biosynthesis), could be involved in the salt tolerance determined by *LCl-6* (Figure 2). The effect of these underlying biosynthesis processes in this QTL region on the differences in root growth under salinity conditions between both hybrids needs to be further investigated. It would not be the first time that DEGs involved in cell wall loosening are associated with reduced root growth in a salt-tolerant citrus cultivar [29]. Additionally, increased cell wall lignification in the root of the salt-tolerant hybrid could render the apoplastic movement of Cl^−^ more difficult.

### 3.3. Salt Tolerance Candidate Genes Underlying LCl-6 in the R×Pr Population

Numerous genes (23) coding for aquaporins and transporters, in general, were detected in *LCl-6* (Table 4). Some of these genes could be involved in the movement of Cl^−^ from root to shoot, as has previously been reported regarding PIP1 and PIP2 coding genes in citrus plants [46,47]. Thus, differences between the genotypes Cleopatra (*C. reshni*), Carrizo (a *C. sinensis* × *P. trifoliata* hybrid), and *P. trifoliata* in PIP1 expression were found to be related to Cl^−^ exclusion from leaves, probably due to the effects on water movement, although salinity did not affect its gene expression [46]. Taking into account the primer sequences provided [46], an isomorph could only be assigned to PIP1 (PIP1.4), corresponding to Ciclev10012384m, which is included in *LCl-6* (Table 4), but with no differential expression between hybrids 107 and 90. In Arabidopsis, PIP2.1 substrate transport activity can be switched between ion and water channel models using phosphorylation [48]. Some NPF proteins have been reported in previous studies to have a major impact on Cl^−^ homeostasis in citrus leaves (NRT1-2; [20] and Arabidopsis roots (NPF2.5; [23]). None of the citrus candidate genes identified by Brumós et al. [21] for Cl^−^ homeostasis are located in *LCl-6* or even in chromosome 6. The rootstock root is the organ of the citrus plant that contributes to the uptake of water and elements from the soil, with these genes coding for transporters belonging to large gene families. The next step, to rank the salt-tolerant candidate genes in the Cleopatra mandarin genome, involved studying root differential gene expression in *LCl-6* using hybrids 90 and 107, two salt tolerance-contrasting full-sibs of the R×Pr population, which share the *Poncirus* allele but differ in relation to the Cleopatra allele in the *LCl-6* region. Thus, 11 genes coding for transporters, putatively involved in ion homeostasis, showed differential expression in *LCl-6*, with *C. clementina* and *P. trifoliata* used as reference genomes (Table 4 and Appendix A, respectively). Four of the seven NPF coding genes located in *LCl-6* exhibited differential expression in at least one of the comparisons carried out: coding genes for NPF5.12 and NPF8.1 in two comparisons and coding for NPF5.9 in three comparisons (Table 4). Noteworthily, Zhao et al. [30] found that salinity induced the differential expression of these NPF coding genes in the roots of *Poncirus trifoliata*, here named Ptrif.0006s0823 (Pt6g013450), Ptrif.0006s2462 (Pt6g014250), Ptrif.0006s0814 (Pt6g013550), and Ptrif.0006s0815 (Pt6g013550) (Appendix A). One member of them, coding for NPF5.12, is the only gene whose expression level in the root is higher in the salt-sensitive genotype than in the salt-tolerant genotype (Appendix A), which could be related to a higher influx of Cl^−^ from soil to root in the salt-sensitive genotype. In Arabidopsis, He et al. [49] found that NPF5.12, which is a vacuolar nitrate efflux transporter, is preferentially expressed in root pericycle cells and xylem parenchyma cells and plays an important role in modulating the allocation of nitrate among roots and shoots. In *Brassica rapa*, NPF5.12 was found to be upregulated in roots under low nitrate conditions, suggesting that it plays a positive role in nitrate absorption [50]. In addition to NPF5.9, another gene coding for a transporter and showing differential expression in most of the comparisons is Ciclev10011745/Ptrif.006s0610, coding for a major facilitator superfamily (MFS) protein (Table 4 and Appendix A). In Arabidopsis, its ortholog regulates salt, drought, heat stress, and turgor-dependent growth through the ABA-dependent signal transduction pathway [51]. The differential expression of these genes has not been validated by qPCR because it is documented [52,53] that results obtained with RNA-seq correlate very consistently with qPCR, and the drastic differences in the expression of genes coding for NPF5.9 and MSF (Appendix A) make them robust enough to not require validation by qPCR or other approaches [54].

The incorporation of relevant candidate genes into the genetic maps improved the resolution of the QTL analyses, particularly for leaf Fe accumulation under salinity conditions in the GP experiment (comparing Appendix A) and root growth under control conditions in the NG experiment (comparing Appendix A); this points to NPF5.9 and the ABCG transporter coding genes as the most appropriate candidate genes, respectively. Thus, the fine mapping of leaf [Fe] in the GP experiment under salinity conditions at exactly NPF5.9 (Appendix A) concurs with the results reported by Chen et al. [55], who found that NPF5.9 and NPF5.8 mediate iron (and nitrate) long-distance transport and homeostasis in *Arabidopsis*.

In the case of leaf Cl^−^ accumulation across the three long-lasting salinity experiments (Figure 3), interval mapping and Kruskal–Wallis analyses point to MFS as the most significant candidate gene regarding Cl*_S (leaf [Cl^−^] in the experiment reported by Raga et al. [17]), leaf Cl^−^ accumulation in the GP experiment (GP_L_Cl_S), and the phenotypic plasticity of leaf [Cl^−^] (dCl_L) in the NG experiment. However, given the shape of the LOD and K profiles, the presence of more than one closely linked candidate gene controlling these leaf Cl-related traits cannot be ruled out, and the ability of MFS and NPF candidate genes to transport Cl^−^ needs to be tested in future experiments. In addition, it must also be noted that phenotypic differences might not be related to differences at the mRNA level but at the protein level (protein abundance, amino acid sequence, etc.).

An important achievement of the present study is the biotechnology developed to speed up and increase the efficiency of rootstock breeding programs in order to confer salt tolerance on citrus varieties as a strategy to mitigate the effects of climate change on the availability of good-quality water. Thus, the markers developed for candidate genes MFS and NPF5.9 facilitate the identification of the Cleopatra allele that increases salt tolerance in rootstock breeding programs, in which Cleopatra has been used as a salt-tolerance donor (Figure 4), thus making marker-assisted selection for this quantitative trait feasible.

## 4. Materials and Methods

### 4.1. Plant Materials

The R×Pr mapping population, consisting of 151 hybrids obtained at IVIA in Spain, is composed of controlled crosses between *Citrus reshni* Hort. ex. Tan. (Cleopatra mandarin) as female parent (salt and iron chlorosis-tolerant and apomictic) and two apomictic and disease-resistant varieties of *Poncirus trifoliata* (L.) Raf. (trifoliate orange, Pr): 83 Flying Dragon hybrids and 68 Rich hybrids as pollinators [10]. Seedlings from mature fruit yielding R×Pr trees were analyzed using molecular markers in order to discard the zygotic seedlings [56] to obtain nucellar plants for the GP (grafted population) and NG (non-grafted) experiments.

The procedures and steps involved in obtaining both mapping and grafted populations were previously described [57]. In the GP experiment (2013–2014), six nucellar seedlings, obtained from each of the 62 R×Pr hybrids, showing apomictic reproduction and parents (Cleopatra and Flying Dragon), were grafted with Clausellina mandarin (*Citrus unshiu* (Mak.) Marc.) and maintained for several years until full production prior to the salinity experiment at IVIA (Valencia, Spain). In the NG experiment (2020), 12 nucellar seedlings, obtained from each of the 42 apomictic R×Pr hybrids and parents (Cleopatra and Rich) in 2018, were grown at the Estación Experimental La Mayora (IHSM-CSIC), Malaga, Spain.

In the GP experiment, two or three trees from the six repetitions (nucellar grafted plants) from each R×Pr hybrid were randomly assigned to the control and salinity treatments from 15 June 2013 to 20 October 2014 (16 months) in a greenhouse (Appendix A, respectively). Plants were grown in pots (17 L) using cocofiber as a substrate. The greenhouse had an automatic roof ventilation and heating system to maintain the interior air temperature above 8 °C. A high-frequency fertirrigation system, together with 4 L h^−1^ drippers, was used and regulated to ensure the homogeneity of the nutrient solution in the roots of all plants cultivated simultaneously. The nutrient solution (pH: 6.4) contained the following concentrations of macronutrients (in mM) NO_3_^−^ 8.1, H_2_PO_4_^−^ 4, SO_4_^2−^ 1, NH_4_^+^ 0.9, K^+^ 4.2, Ca^2+^ 3.5, and Mg^2+^ 1, in addition to the following concentrations of micronutrients (in µM): Mn^2+^ 8, Zn^2+^ 2.3, B 20, Cu^2+^ 7, Mo^4+^ 0.5, and Fe^2+^ 15.3. The water for the nutrient solution was previously subjected to reverse osmosis treatment. Although the salinity-treated plants were similarly irrigated, the nutrient solution was supplemented with 30 mM NaCl.

In the NG experiment (Appendix A), six plants from the twelve repetitions (eight-month-old nucellar seedlings) of each apomictic R×Pr hybrid and parent were randomly selected to be subjected to control and salinity treatments over a period of a year from 1 October 2019 to 1 October 2020 in a greenhouse under natural light conditions with no temperature control. Plants were grown under natural greenhouse conditions in pots (4 L) using a vermiculite substrate. The nutrient solution (electrical conductivity:1.68, pH: 7.13) contained the following concentrations of macronutrients (in mM) NO_3_^−^ 8.45, H_2_PO_4_^−^ 0.74, SO_4_^2−^ 1.84, K^+^ 6.08, Ca^2+^ 4.25, and Mg^2+^ 1.33, in addition to the following concentrations of micronutrients (in µM) Fe^2+^ 66, Mn^2+^ 1.33, Zn^2+^ 2.3, B 17, and Cu^2+^ 2.3, supplemented with either 0 mM NaCl for control (1.67 dS m^−1^) or 15 mM NaCl for the saline treatment (4.67 dS m^−1^) until completion of the experiment. Plants were watered automatically using 2.5 L/h drippers three times per week, on alternate days, receiving 200 mL at each irrigation event.

### 4.2. Trait Evaluation

For the GP experiment, several physiological (Na, Ca, K, Fe, and Cl concentrations in mature leaves) and agronomic (related to fruit yield and quality) traits of the grafted satsuma variety were evaluated (see Appendix A for abbreviations) under both control and salinity conditions, denoted by the suffixes_C and _S, respectively. Three fully developed leaves per plant were sampled from vegetative spring shoots after 15 months of treatment in order to measure total chlorophyll leaf concentration (TChl, µmol m^−2^) using an SPAD-502 Plus chlorophyll meter (Konica Minolta Inc., Tokyo, Japan) and the function TChl = [(0.2861×SPAD value) − 6.9501]. Dry tissue samples of these leaves, maintained at 80 °C for 3 days, were prepared for mineral analysis by digestion in a HNO_3_:HClO_4_ (2:1, *v*/*v*) solution. Inorganic solutes were determined in parts per million (ppm) (mg Kg^−1^) using the ICAP 6500 DUO/IRIS Intrepid II XDL spectrometer at the Segura Center for Soil Science and Applied Biology ionomics facility (CEBAS-CSIC; Murcia, Spain). Foliar concentrations of Cl^−^ (milligrams per liter) were also evaluated using a chloride analyzer (Model 926, Sherwood Scientific, Cambridge, UK) and the methodology described by Gilliam [58].

Fruit yield was evaluated in terms of the number of normal, ripe fruit (FN) and total fruit weight (TFW, in g). Original trait FN was analyzed using a Poisson distribution (FNp) for QTL analysis. A minimum of five randomly sampled fruits per tree were also evaluated for the following internal fruit-quality traits at two harvest times (1: mid-September 2014; 2: mid-October 2014): fruit weight (FW, in grams), juice titratable acidity (A, %), and soluble-solids content (SSC, as ° Brix) using a Pallete PR-101 digital refractometer (Atago, Tokyo, Japan).

At the end of the NG experiment, after one year of treatment, 8–10 leaves from each plant were harvested and dried to determine Cl, Na, K, Ca, and Fe, as described for the GP experiment (see Appendix A for abbreviations). Total C and N leaf concentrations (g/100 g of dry weight) were also determined using a Leco TruSpec CN628 elemental analyzer at the ionomics facility (CEBAS-CSIC). The roots of each plant were exhaustively washed with tap water and dried. Total root dry weight (TRDW, in grams) was estimated for each plant. Fine-root mass per plant was also evaluated by weight (FRDW, in grams). Cl, Na, K, Ca, Fe, C, and N were determined in the fine roots of each plant. To characterize the distribution of Cl and, similarly, of Ca in the plant, the function ([Cl]_root_ − [Cl]_leaf_)/[Cl]_root_ was calculated and used as an additional trait (ClR-ClL/ClR) at both salinity levels. The phenotypic plasticity of leaf and root Cl concentrations (dCl_L and dCl_R, respectively) and other traits were estimated as the difference between salinity and control means relative to the mean under control conditions [59]. The responses of genotypes to salinity, also viewed as phenotypic plasticity, regarding chloride-related traits and root mass, were depicted as reaction norms [60].

### 4.3. Statistical Analysis

The GP experiment used a split-plot design with four blocks, using NaCl treatments as the main plot and rootstocks as the subplots. With regard to the statistical analysis of the experiment, the blocks were random. On the other hand, to study the genotype (G) and salinity (E) effects, as well as the G×E interactions of the evaluated traits, the effects of genotype and treatment were classed as fixed. In the NG experiment, genotypes were distributed at random within each NaCl treatment. Considering R×Pr hybrid genotypes as a random effect factor, broad-sense heritability (H^2^) was estimated for all traits with respect to nucellar rootstocks (repetitions) derived from apomictic R×Pr hybrids under control or salinity conditions based on the genotypic (V_G_) and environmental (V_E_) variance estimators calculated using the minimum variance quadratic unbiased estimator (MIVQUE), as described elsewhere [61].

Spearman’s correlation coefficients were used to study the relationships between the different traits.

### 4.4. Genetic Analyses

#### 4.4.1. Linkage Map and QTL Analyses

QTL analyses were carried out using the genotypic and map data from [10] based on SSR, IRAP, and SCAR markers, as well as the adjusted means of traits. The interval mapping (IM) procedure and the Kruskal–Wallis rank sum test (nonparametric QTL approach), together with the MapQTL^®^ 6 program [62], were used to identify QTLs. QTL analyses were carried out in two different ways. Firstly, we analyzed the data as a cross-pollinated (CP) population type in order to examine intralocus interactions. Secondly, using a two-way pseudo-test cross approach, we analyzed data for each parental meiosis separately [63]. Although this second approach takes advantage of the computational benefits of the two-genotype QTL model, intralocus interactions were ignored, rendering this approach less powerful and realistic [62]. JoinMap 4.1 software [64] was used to translate and split the marker data in order to separate the two meiosis. Being parent-specific, some linkage groups or linkage group parts (R9a, R6, R4a, Pr1, Pr4a, and Pr9b) were ignored when using the CP data for QTL analysis. The Cleopatra map contains 86 markers distributed among 10 linkage groups, covering 1127.127 cM of the *C. reshni* genome. Similarly, the *Poncirus* map contains 73 markers distributed among 11 linkage groups, covering 1416.759 cM of the *Poncirus trifoliata* genome. No genotypic differences between the male parents, Flying Dragon and Rich, were detected with respect to these mapped markers, which is in line with the high degree of relatedness found among the *P. trifoliata* accessions [34,65]. The CP map contains 93 markers, spread among nine linkage groups, covering 1406.761 cM of the integrated genome.

With respect to IM, experiment-wise significance was assessed to be 5% using 1000 permutation tests. These LOD critical values ranged from 1.1 to 2.0, depending on the specific trait and linkage group in the two-way pseudo-testcross analysis (population type DH). On the other hand, the critical LOD values ranged from 2.1 to 3.4, depending on the specific trait and linkage group in the CP analysis. Only significant QTLs with LOD ≥ 2.36 for heritable traits (H^2^ > 0) are reported here. With regard to the Kruskal–Wallis procedure, the significance level for individual tests was fixed at 0.005, as recommended in the manual [62].

#### 4.4.2. Candidate Genes and Linkage Analysis

A genomic region around the CR23,750 marker on the integrated linkage group 4c was found to be particularly rich in QTLs and to have QTLs for Cl-related traits. With respect to this region, markers from the CP map were anchored to the physical map of *C. clementina* using primer and/or EST sequences, as well as the BLASTN tool (https://phytozome.jgi.doe.gov/pz/portal.html#!search?show=BLAST accessed on 18 February 2019). All genes from marker 15R,750 (at Ciclev10011720m.g) down to marker CR28,270 (at Ciclev10011175m.g) on integrated linkage group 4c were downloaded from both the *C. clementina* and *Poncirus trifoliata* genomes at https://phytozome.jgi.doe.gov. *C. clementina* was chosen instead of *C. sinensis* because *C. reshni* (Cleopatra mandarin) is genetically closer to *C. clementina* than to *C. sinensis* [66,67]. The annotation of some genes downloaded from https://phytozome.jgi.doe.gov (accessed on 18 February 2019) was tested by blasting their peptide sequence at https://ncbi.nlm.nih.gov (accessed on 18 February 2019). Gene ontology (GO) enrichment analysis of genes was carried out using the Singular Enrichment Analysis tool [68] on the AgriGo platform (http://systemsbiology.cau.edu.cn/agriGOv2/ accessed on 20 February 2019).

Aligned DNA sequences of relevant candidate genes were used to find divergent regions in order to design primers that reveal insertion/deletion (InDel) polymorphisms in the R×Pr population. Forward and reverse primers covering a divergent region for each candidate gene were designed using https://bioinfo.ut.ee/primer3/ (accessed on 20 February 2019) in order to develop sequence-characterized amplified region (SCAR) markers in or near these genes to examine their genetic location and quantitative effects on traits (Appendix A). Genomic DNA extractions from leaf tissue were carried out for each hybrid. Polymerase chain reaction (PCR) conditions were specific to each marker, and the resulting product was analyzed by electrophoresis in 10% DNA sequencing-type polyacrylamide gels and was revealed by silver staining. All procedures are described elsewhere [56]. PCR products from homozygous plants were sequenced at the López-Neyra Institute of Parasitology and Medicine DNA sequencing and genomics facility (Granada, Spain) to test their gene identification/physical location (Appendix A).

JoinMap 4.1 mapping software [64] was used for segregation and linkage analysis of candidate genes in the R×Pr population (151 hybrids) using a linkage criterion LOD ≥ 10, a recombination fraction of 0.5, and the Kosambi mapping function. The CP population type, resulting from a cross between two heterogeneously heterozygous and homozygous diploid parents, was selected to analyze R×Pr progeny with no previous knowledge of the marker linkage phase. This strategy uses nn × np or lm × ll codifications for dominant markers that segregated into either *P. trifoliata* or *C. reshni*, respectively, in addition to ab × cd, ef × eg, and hk × hk codifications for codominant marker loci, at which 4, 3, and 2 different alleles are segregating, respectively.

### 4.5. DNA Sequencing of Selected Materials

For DNA and RNA sequencing, we selected two full-sibs of the R×Pr population, which differed in the Cleopatra allele at marker CR23,750, and the degree of salt tolerance measured in terms of both fruit yield and leaf [Cl^−^] under salinity conditions in the GP experiment. These hybrids, which share the male parent Flying Dragon at CR23,750, were: hybrid 107 (salt tolerant with an ac genotype at QTL marker CR23,750) and hybrid 90 (salt sensitive with a bc genotype at QTL marker CR23,750). Additionally, using marker analysis, a homozygote (bb) at CR23,750 (22-7) was selected from a progeny obtained by self-pollination from monoembryonic hybrid 22, which belongs to the R×Pr mapping population and was also derived from Flying Dragon. DNA extractions, sequencing, and bioinformatic sequence analysis were carried out by Sistemas Genomicos S.L. (Paterna, Valencia, Spain). Leaves (around 5 g) from the 107, 90, and 22-7 trees were used for DNA extraction with the aid of a DNeasy Plant Mini Kit (Qiagen, Barcelona, Spain). Three DNA libraries were generated using the NEBNext DNA Library Prep kit for sequencing on the Illumina HiSeq platform, with an expected output per sample of 40 Gb (2 × 150 pb, single index, 100× coverage). The quality control of initial reads was analyzed using the *FastQC* method. The reads were then mapped against the *Citrus clementina* genome (GCF_000493195.1). The low-quality mapping reads were filtered using the *samtools* method [69], and the PCR duplicates were eliminated with the aid of *Picard* tools.

The reads of candidate genes in the salt tolerance QTL were used to perform de novo assembly of alleles (a, b, and c/d) at each locus based on three steps: (1) mapping of initial reads to the whole genome using the *bwa* algorithm [70] and discarding low-quality reads and duplicated PCR products through post-processing alignment; (2) selection of reads mapped to the candidate genes in the QTL, sequence extraction to perform de novo assembly; and (3) haplotype identification using the *WhatsHap* algorithm [71] with default parameters. The sequences of candidate alleles were obtained using the haplotypes identified in the previous step and through multiple alignments with the aid of the *Muscle* program [72].

### 4.6. RNA-Sequencing Analysis of the Root of Selected Materials

#### 4.6.1. RNA Isolation of Selected Materials

The same two full sibs of the R×Pr population, hybrids 107 and 90, were selected for the root transcriptomic study at the end of the NG experiment. After 383 days of salt treatment, the whole root of five plants per genotype and treatment was taken and maintained with liquid nitrogen for further RNA extraction and to estimate Cl, Na, K, Ca, Fe, C, and N concentrations.

Total RNA from the whole root was purified as described elsewhere [73]. Briefly, three independent biological samples, with one plant per sample and treatment, were used for analysis in an initial purification step with CTAB extraction buffer, followed by separation in chloroform:isoamyl alcohol (24:1, *v*:*v*) and LiCl precipitation [74]. The resuspended RNA was further purified using the Aurum^TM^ total RNA mini kit (Bio-Rad Laboratories S.A., Madrid, Spain), together with RNAse-free DNase in-column treatment (Promega Biotech Ibérica SL) according to the manufacturer’s instructions. RNA quantity, quality, and integrity were determined based on absorbance ratios at 260 nm/280 and 260 nm/230 nm, using a mySPEC NanoDrop spectrophotometer (VWR International bvba, Leuven, Belgium) and agarose gel electrophoresis and further verified with the aid of a 2100 Bioanalyzer system (Agilent Technology, Madrid, Spain) at the López-Neyra Institute of Parasitology and Biomedicine (IPBLN-CSIC, Granada, Spain) genomic facility. Sample RIN values ranged from 5.9 to 8.2.

#### 4.6.2. Library Preparation and Sequencing

The cDNA libraries were constructed and sequenced using pair-end sequencing (2 × 74 bp) with the aid of the Illumina NextSeq 500 System MidOutput 150-cycle at the IPBLN-CSIC genomic facility (Granada, Spain). A total of 298 million reads, ranging from 21.8 million in NGS083-20-4 to 26.1 million in NGS047-20-1, were generated and filtered for high-quality reads (Q30, 91.62%). Raw reads were deposited under the name Bioproject PRJEB61142.

#### 4.6.3. Read Pre-Processing and Counting

Raw reads were analyzed using *FastQC/MultiQC* v1.11 software [75] to verify their suitability and were pre-processed using the *SeqTrimBB* v2.1.8 program (https://github.com/rafnunser/seqtrimbb, accessed on 15 March 2022). The standard parameters for Illumina paired reads (minimal length of 40 bases and contaminant minimal ratio of 0.65) were used. The resulting clean reads from every sample were mapped to the transcriptomes of the *C. clementina* genome v1.0 (33,929 protein-coding transcripts; https://phytozome-next.jgi.doe.gov/info/Cclementina_v1_0, accessed on 15 March 2022) and the *Poncirus trifoliata* genome v1.3.1 (33,229 protein-coding transcripts; https://phytozome-next.jgi.doe.gov/info/Ptrifoliata_v1_3_1 accessed on 15 March 2022) using *Bowtie* v2.4.4 [76] with default parameters and *–no-unal –no-mixed –no-discordant* options to retain only appropriately mapped paired-end reads in the Sequence Alignment/Map (SAM) file. SAM file sorting and indexing were conducted using SAM tools [69]. Read counting per transcript was accomplished using *sam2counts* (https://github.com/vsbuffalo/sam2counts accessed on 15 March 2022) to obtain a table in which the rows were transcripts and the columns were samples.

#### 4.6.4. Differential Expression, Correlation, and Clustering Analyses

Transcript counts were submitted to our pipeline *RNAseqFlow* [77] for (1) filtering (removal of transcripts with <1 count per million [CPM] reads in >9 samples); (2) normalization using the trimmed mean of M-values (TMM) method implemented in the *edgeR* v3.36.0 library [78]; (3) heteroskedasticity removal using the *limma::voom()* function [79] to facilitate reliable linear adjustment; (4) under a negative binomial distribution, fitting a generalized linear model with the *limma* v3.50.3 library; as in the present study, with its small number of replicates, its power advantage may outweigh the possibly exaggerated number of false positives [80]; (5) differential expression using the *limma::treat()* function [81] to enhance the biological implications of the results [79] and to prevent the misuse of statistical significance [82,83]; and (6) correlation-based clustering and further definition of communities using the *igraph* v1.3.5 library in order to detect gene co-expression networks [84,85,86]. A differentially expressed gene (DEG) has at least an adjusted *p*-value < 0.1 and an absolute fold change |FC| > 1.2, as indicated by [81]. Minimal Pearson correlation was set to 0.75 [87]. Those genes being DEG in all contrasts or having a *Kleinberg’s centrality score* > 0.9 [88] and a degree >10 were considered for further analyses.

#### 4.6.5. Functional Analyses

Functional interpretations based on gene ontology (GO) and KEGG pathways were conducted using the closest *Arabidopsis* ortholog to the *Citrus* or *Poncirus* transcript. Orthologs were obtained using the *Arabidopsis thaliana* TAIR10 proteome from ENSEMBL and a reciprocal comparison based on the fast and sensitive protein aligner DIAMOND v.2.0.14 [89] using default parameters and –*max-target-seqs 1* in –*ultra-sensitive* mode to obtain the best hit. The same protocol was used to obtain the orthology between the *C. clementina* v1.0 and *Poncirus trifoliata* v1.3.1 transcripts. REVIGO [90] was used for a list of GOs and KEGGs in which genes are involved. AgriGO v2.0 [68] and ShinyGO 0.76 [91] were used for functional enrichment with the aid of the background for *A. thaliana*, running on default parameters (in all cases, FDR < 0.05).

## 5. Conclusions

In this original approach to determine genes conferring salt tolerance in citrus, we describe the integration of QTL and RNA-seq analyses to identify transporter coding genes as likely involved in the leaf chloride exclusion mechanism of the salt-tolerant Cleopatra mandarin. Thus, we showed that the salt tolerance of Cleopatra mandarin rootstock is inherited by its R×Pr progeny, which is consistently controlled by at least one QTL on chromosome 6 (*LCl-6*) across experiments using both grafted and non-grafted apomictic progenies thus pointing to the root as the main organ involved. The study of differential gene expression at the root of two salt tolerance-contrasting full-sibs, which differ in relation to the Cleopatra allele in *LCl-6*, has identified some transporter coding genes in this QTL as positional candidate genes to explain xylem chloride exclusion under salinity conditions, as well as other genes that could be related to the growth and lignification of the root; this suggests that other salt tolerance mechanisms involved in and underlying salt tolerance in the same genomic region may be present. Markers developed for candidate genes in *LCl-6* constitute new tools to increase the efficiency and speed of salt tolerance breeding programs for citrus rootstocks and to set up core collections of *Citrus* and *Poncirus* accessions where new salt-tolerant materials might be more easily discovered.

## Figures and Tables

**Figure 1 ijms-24-15759-f001:**
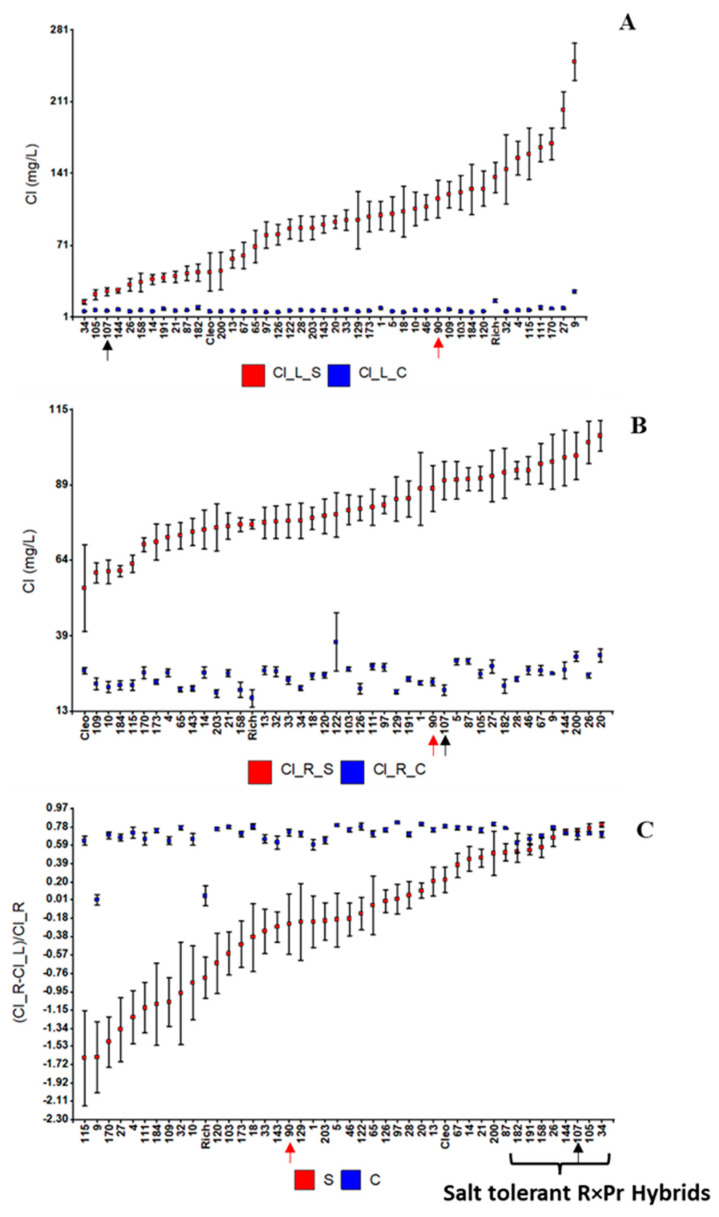
Means and standard deviations of R×Pr hybrids and parents (Cleopatra, Cleo, and trifoliata Orange, Rich) under both control and salinity conditions for (**A**) leaf [Cl^−^], (**B**) root [Cl^−^], and (**C**) their difference relative to root [Cl^−^]. The salt-tolerant (107) and salt-sensitive (90) hybrids are indicated by an arrow (black and red, respectively).

**Figure 2 ijms-24-15759-f002:**
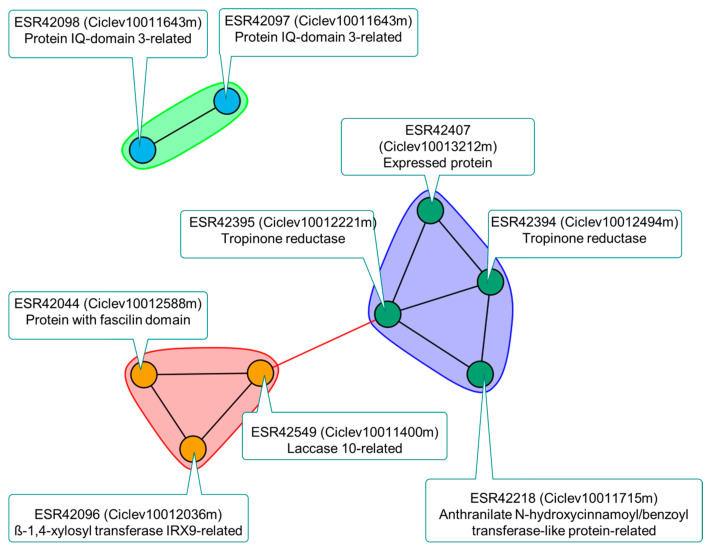
Co-expression network for differentially expressed genes (DEGs) in AHC-1 (Appendix A) in the salt tolerance quantitative trait locus (QTL) *LCl-6*. Names of transcripts and genes are provided, together with their putative function, showing that processes related to cell wall biosynthesis (specifically, glucuronoxylan and xylan biosynthesis, as well as lignin metabolism) and secondary metabolism (alkaloid/phytoalexin biosynthesis) at the root could be involved in salt tolerance.

**Figure 3 ijms-24-15759-f003:**
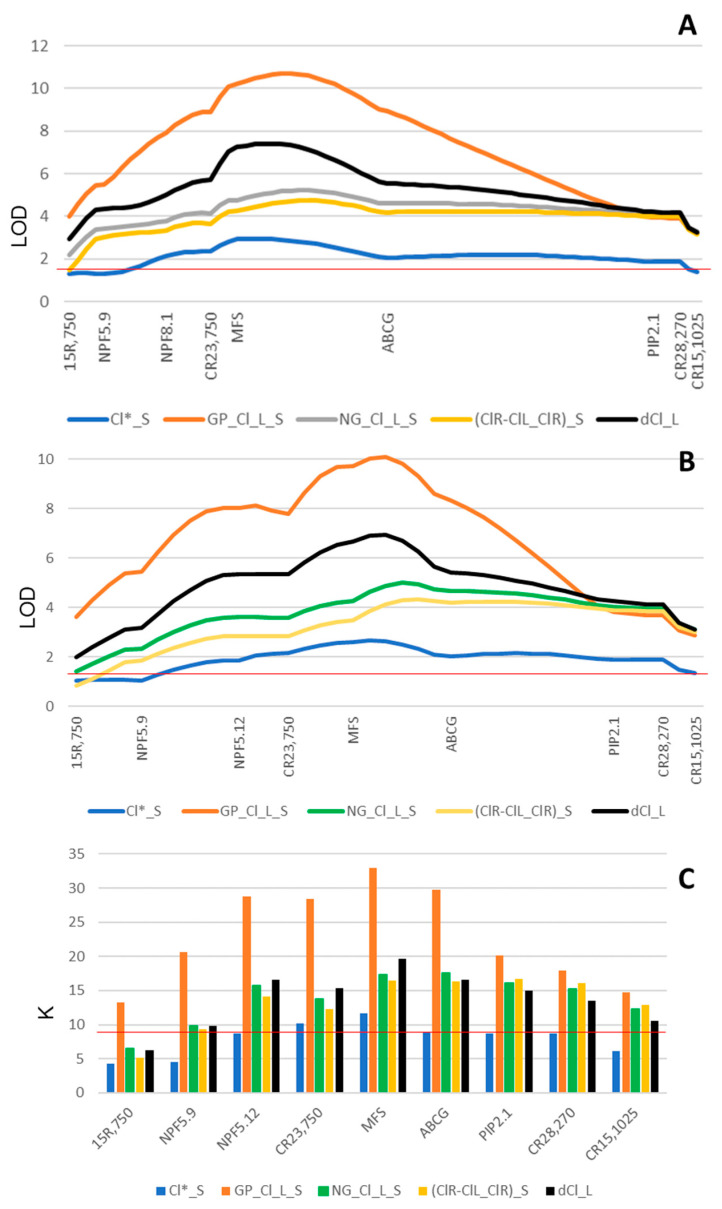
Log of odd ratio (LOD) profiles of quantitative trait loci (QTLs) for Cl-related traits in linkage group 4c (**A**) and R4c (**B**) after incorporating candidate genes into the linkage maps and using Kruskal–Wallis statistic K for the association between phenotype and Cleopatra marker/gene segregation (**C**). The red horizontal line indicates significant levels. Cl*_S corresponds to the salt tolerance QTL reported by [17]. To distinguish between experiments, the prefix GP or NG has been added to the trait name (Cl_L_S).

**Figure 4 ijms-24-15759-f004:**
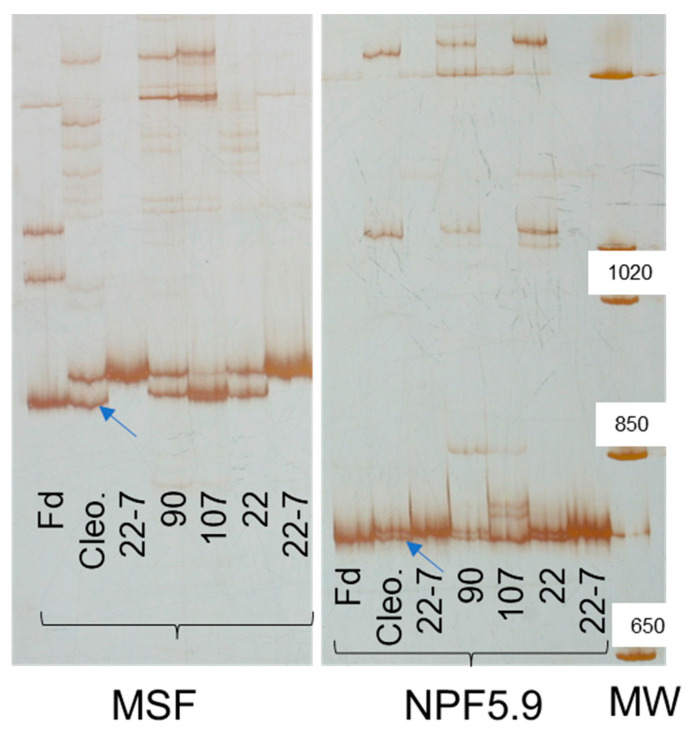
Sequence-characterized amplified region (SCAR) markers (MFS and NPF5.9) revealed by silver-stained polyacrylamide gel electrophoresis in Flying Dragon (Fd, trifoliate orange), Cleopatra (Cleo.), 3 R×Pr hybrids (90, 107 and 22), and homozygous line 22-7. Blue arrows indicate the beneficial allele (the fastest/smallest Cleopatra band in both cases) present in hybrid 107. The MW Lane corresponds to the molecular weight marker (sizes in bp).

**Table 1 ijms-24-15759-t001:** List of positions in centiMorgan (cM), log of odd ratio (LODs,) as well as nearest markers or marker intervals of quantitative trait loci (QTLs) detected by IM in the GP experiment in the integrated *Citrus reshni-Poncirus trifoliata* genetic linkage map (LG) using the cross-pollinated model. The QTLs that were also detected in the individual parental linkage maps are indicated by putting the parental linkage groups in parentheses (R and Pr for *C. reshni* and *P. trifoliata,* respectively). The four genotypic means (**ac**, **ad**, **bc**, and **bd**, being *C. reshni*
**ab** and *P. trifoliata*
**cd**), as well as the percentage of explained variance (PEV), are also indicated. LOD values higher than genome-wide significant LOD scores are indicated in bold.

Trait	Group	Position	Marker	LOD	ac	ad	bc	bd	PEV
Ca_L_S	4c (R)	0.00	15R,750	3.02	25,445.20	36,035.50	23,826.60	26,975.70	21.60
Cl_L_S	3b	15.08	C2iC1i,470	2.80	62.10	55.40	81.08	45.74	20.20
Cl_L_S	3b (Pr)	32.51	TAA27,235	3.81	60.53	51.58	83.46	54.24	26.50
Cl_L_S	4c (R)	9.96	CR23,750	**9.08**	38.82	47.32	90.37	76.14	52.00
Cl_L_S	3a	34.57	CR31,100	2.42	80.54	64.20	40.57	85.82	17.80
Fe_L_C	3b	17.23	5F4R,600	2.62	77.48	69.74	65.80	79.66	19.10
Fe_L_S	4c (R)	6.00	15R,750-CR23,750	2.60	71.56	74.68	54.50	63.11	19.00
FW1_S	12 (R)	45.86	CMS20,170-6F5R,1200	2.88	87.90	91.06	66.38	65.28	22.10
K_L_S	4c (R)	18.96	CR23,750-CR28,270	2.56	10,614.20	10,767.70	16,442.40	14,337.00	18.70
NFp_C	10+5b (R)	175.19	CMS46,190	2.88	10.51	11.75	21.28	12.14	20.80
NFp_C	4c (R)	9.96	CR23,750	2.48	17.07	17.20	9.39	12.26	18.10
NFp_S	3b (Pr)	6.98	CR71,310	**4.25**	10.52	11.93	6.54	18.84	29.10
NFp_S	4c (R)	19.96	CR23,750-CR28,270	**5.18**	15.72	13.46	5.11	8.09	34.20
SSC1_C	4c (R)	14.96	CR23,750-CR28,270	2.39	8.89	9.08	8.23	8.63	18.70
SSC1_S	12 (R)	25.84	CHI_M598-6F5R,1200	2.95	8.57	8.76	9.66	9.24	22.60
SSC1_S	4c (R)	18.96	CR23,750-CR28,270	7.10	9.77	9.63	8.06	9.05	46.00
SSC2_C	4b (R)	90.52	520AR,350-Py65C,506	3.52	8.46	8.68	9.36	7.96	26.80
SSC2_S	4c (R)	27.78	CR28,270-CR15,1025	3.05	9.26	9.55	8.53	8.67	23.60
TFW_C	4b (Pr)	30.43	CR72,260	3.26	782.95	1342.33	1056.52	1392.10	23.20
TFW_C	3b (Pr)	17.23	5F4R,600	2.85	1181.72	1420.13	911.87	1655.12	20.60
TFW_C	4c (R)	9.96	CR23,750	**4.08**	1481.83	1429.62	719.21	1098.06	28.10
TFW_S	3b (Pr)	8.90	C8iC1rt,650	3.42	769.45	922.67	512.03	1182.29	24.10
TFW_S	4c (R)	14.96	CR23,750-CR28,270	**5.61**	1115.86	983.97	387.39	589.08	36.50

**Table 2 ijms-24-15759-t002:** Means and standard errors for parents Cleopatra, trifoliate orange, whole population (Pop) and the salt-sensitive (SS) and tolerant (ST), reference hybrids (90 and 107, respectively) depending on the experiment and treatment (Exp/E). Trait codes are summarized in Appendix A for GP and NG experiments, respectively.

Exp/E	Trait	Cleopatra	Trifoliate	90 (SS)	107 (ST)	Pop
	A1	1.3 ± 0.0	1.8 ± 0.2	1.1 ± 0.0	1.1 ± 0.1	1.3 ± 0.0
	A2	0.8 ± 0.0	1.0 ± 0.1	0.9 ± 0.1	0.8 ± 0.0	0.9 ± 0.0
	Ca_L	28,639.1 ± 5688.0	30,049.4 ± 4049.4	17,460.5 ± 3035.6	33,967.9 ± 9743.0	28,109.6 ± 619.6
	Cl_L	10.2 ± 1.3	20.7 ± 3.5	17.5 ± 7.5	12.8 ± 2.3	13.3 ± 0.4
	Fe_L	55.8 ± 2.4	103.5 ± 6.0	82.7 ± 50.2	62.7 ± 8.0	72.6 ± 1.7
	FW1	97.8 ± 21.9	53.5 ± 15.9	77.3 ± 18.0	122.8 ± 50.1	89.7 ± 3.3
GP/C	FW2	63.9 ± 0.0	71.1 ± 16.3	79.5 ± 24.6	158.1 ± 70.3	105.6 ± 3.7
	K_L	17,095.0 ± 2611.4	12,537.4 ± 1676.2	14,715.9 ± 5314.0	14,645.5 ± 2101.2	16,843.6 ± 378.1
	Na_L	1111.6 ± 439.0	635.6 ± 106.5	498.0 ± 193.8	641.8 ± 152.09	665.3 ± 34.2
	NFp	6.7 ± 3.3	11.7 ± 2.3	27.5 ± 8.5	10.0 ± 7.0	13.7 ± 0.9
	SSC1	8.5 ± 0.3	9.7 ± 0.1	8.2 ± 1.1	8.1 ± 0.2	8.7 ± 0.1
	SSC2	7.2 ± 0.0	9.8 ± 0.8	8.6 ± 0.9	7.5 ± 0.0	8.7 ± 0.1
	TChl	12.8 ± 0.3	14.6 ± 0.6	15.6 ± 0.9	15.2 ± 1.7	14.5 ± 0.1
	TFW	545.8 ± 172.5	672.6 ± 75.5	1894.6 ± 252.6	1006.3 ± 432.2	1166.9 ± 64.1
	A1	1.4 ± 0.3	1.1 ± 0.0	1.1 ± 0.0	1.1 ± 0.1	1.5 ± 0.1
	A2	0.9 ± 0.2	0.9 ± 0.1	0.6 ± 0.0	0.9 ± 0.1	1.0 ± 0.0
	Ca_L	23,834.1 ± 2561.0	17,674.6 ± 339.9	29,286.7 ± 7462.5	19,806.7 ± 2339.4	28,302.8 ± 996.4
	Cl_L	74.0 ± 7.1	110.0 ± 21.6	72.3 ± 2.3	31.3 ± 1.4	66.0 ± 3.5
	Fe_L	52.6 ± 2.3	48.7 ± 2.8	86.6 ± 12.2	83.7 ± 27.1	65.6 ± 2.0
	FW1	76.0 ± 14.4	91.5 ± 21.1	134.4 ± 58.6	83.6 ± 9.1	75.5 ± 3.2
GP/S	FW2	112.3 ± 27.9	59.1 ± 3.1	88.3 ± 0.0	91.2 ± 12.7	82.5 ± 3.4
	K_L	13,709.9 ± 2305.3	21,367.5 ± 1700.9	16,603.8 ± 4939.4	13,042.6 ± 1501.2	13,213.8 ± 601.1
	Na_L	8020.5 ± 348.5	4091.0 ± 179.3	2541.9 ± 268.4	3162.0 ± 348.4	3910.2 ± 181.9
	NFp	5.7 ± 2.2	5.7 ± 2.7	7.0 ± 6.0	15.0 ± 5.0	10.1 ± 0.8
	SSC1	9.6 ± 0.2	8.3 ± 0.3	7.9 ± 0.6	9.4 ± 0.4	9.1 ± 0.1
	SSC2	9.1 ± 0.4	7.7 ± 0.4	7.8 ± 0.0	8.5 ± 0.4	8.9 ± 0.1
	TChl	11.4 ± 2.1	10.5 ± 2.9	15.3 ± 0.1	15.5 ± 0.7	13.8 ± 0.3
	TFW	434.6 ± 54.5	407.6 ± 133.1	670.0 ± 477.0	1352.8 ± 413.3	729.0 ± 51.7
	C_L	39.4 ± 0.9	41.8 ± 0.7	36.2 ± 1.5	36.1 ± 0.5	39.0 ± 0.2
	C_R	44.4 ± 0.0	42.1 ± 0.7	43.4 ± 0.7	43.3 ± 0.2	43.1 ± 0.2
	Ca_L	23,932.3 ± 2685.1	20,284.0 ± 1305.0	26,355.3 ± 1455.1	20,183.0 ± 216.0	22,202.5 ± 461.4
	Ca_R	8955.0 ± 265.1	9642.0 ± 893.2	9293.5 ± 197.5	8797.0 ± 927.0	10,153.9 ± 290.9
	Cl_L	5.8 ± 0.4	16.3 ± 1.9	7.0 ± 0.7	6.6 ± 0.5	7.4 ± 0.5
	Cl_R	26.8 ± 1.0	17.7 ± 2.9	23.1 ± 1.3	20.2 ± 1.6	25.1 ± 0.6
	ClR-ClL/ClR	0.8 ± 0.0	0.1 ± 0.1	0.7 ± 0.0	0.7 ± 0.0	0.7 ± 0.0
NG/C	Fe_L	153.3 ± 6.5	166.3 ± 10.5	861.7 ± 636.3	152.3 ± 30.7	205.6 ± 21.3
	Fe_R	900.3 ± 129.8	2225.3 ± 562.0	928.0 ± 113.0	985.5 ± 458.5	1321.2 ± 163.7
	FRDW	4.0 ± 0.5	6.7 ± 2.1	4.3 ± 1.3	3.5 ± 0.3	6.3 ± 0.3
	K_L	41,733.3 ± 1295.0	31,028.0 ± 1691.3	48,567.3 ± 3066.9	57,188.3 ± 1655.2	48,037.6 ± 1198.6
	K_R	18,175.7 ± 1315.7	26,780.7 ± 1087.7	31,479.5 ± 3552.5	18,780.0 ± 2566.0	22,892.4 ± 539.7
	N_L	3.2 ± 0.2	2.7 ± 0.3	3.9 ± 0.9	4.4 ± 0.6	3.3 ± 0.1
	N_R	1.8 ± 0.0	1.9 ± 0.1	2.3 ± 0.3	1.9 ± 0.1	2.2 ± 0.0
	Na_L	1767.0 ± 154.4	360.3 ± 139.4	946.3 ± 330.3	817.3 ± 109.5	1073.0 ± 91.5
	Na_R	1142.0 ± 196.7	381.7 ± 86.5	960.0 ± 134.0	290.5 ± 15.5	864.7 ± 56.8
	TRDW	15.5 ± 0.7	17.5 ± 3.6	15.6 ± 2.4	14.7 ± 0.3	20.0 ± 0.6
	C_L	42.8 ± 0.3	37.8 ± 1.2	36.7 ± 1.1	39.2 ± 0.4	39.1 ± 0.2
	C_R	44.5 ± 0.3	43.0 ± 0.3	43.6 ± 0.3	43.3 ± 0.2	43.2 ± 0.1
	Ca_L	19,948.0 ± 772.0	19,171.3 ± 726.3	24,257.0 ± 429.1	14,560.0 ± 1743.0	22,202.5 ± 461.4
	Ca_R	6161.5 ± 697.5	7710.7 ± 146.0	7839.5 ± 68.5	7969.0 ± 588.0	7498.5 ± 122.2
	Cl_L	44.5 ± 18.5	137.0 ± 15.0	116.0 ± 18.4	25.4 ± 3.8	92.0 ± 8.0
	Cl_R	54.5 ± 14.5	76.0 ± 1.5	88.0 ± 7.8	90.8 ± 6.3	82.2 ± 1.8
	ClR-ClL/ClR	0.2 ± 0.1	−0.8 ± 0.2	−0.2 ± 0.3	0.7 ± 0.0	−0.2 ± 0.1
NG/S	Fe_L	148.5 ± 12.5	1029.7 ± 806.2	1154.3 ± 947.0	162.3 ± 17.3	261.8 ± 27.5
	Fe_R	534.0 ± 89.0	539.3 ± 175.4	905.5 ± 114.5	1031.5 ± 30.5	835.7 ± 50.5
	FRDW	8.8 ± 2.9	3.7 ± 1.0	10.2 ± 0.1	4.7 ± 0.7	7.5 ± 0.4
	K_L	29,310.5 ± 3005.5	43,560.7 ± 2852.0	44,553.3 ± 1380.2	41,241.7 ± 939.9	42,725.8 ± 900.6
	K_R	11,289.5 ± 1351.5	27,783.3 ± 1288.0	19,544.5 ± 471.5	22,475.5 ± 1169.5	19,709.1 ± 482.1
	N_L	2.8 ± 0.4	3.0 ± 0.4	2.5 ± 0.0	3.2 ± 0.2	2.9 ± 0.1
	N_R	1.7 ± 0.1	1.8 ± 0.1	1.9 ± 0.1	1.8 ± 0.2	1.9 ± 0.0
	Na_L	10,553.0 ± 377.0	8323.3 ± 1149.8	6783.3 ± 881.0	3609.7 ± 579.2	7082.8 ± 474.7
	Na_R	3252.0 ± 1838.0	2518.0 ± 193.4	4180.5 ± 474.5	3012.0 ± 734.0	3490.3 ± 167.1
	TRDW	20.6 ± 3.1	13.5 ± 1.7	22.9 ± 0.2	16.2 ± 0.7	21.0 ± 0.6

**Table 3 ijms-24-15759-t003:** List of positions in centiMorgan (cM), log of odd ratio (LODs), nearest markers, or marker intervals of quantitative trait loci (QTLs) detected by IM in the NG experiment in the integrated *Citrus reshni-Poncirus trifoliata* genetic linkage map (LG) using the cross-pollination model. The QTLs that were also detected in the individual parental linkage maps are indicated by putting the parental linkage group in parentheses (R and Pr for *C. reshni* and *P. trifoliata,* respectively). The four genotypic means (**ac**, **ad**, **bc**, and **bd**, being *C. reshni* **ab** and *P. trifoliata* **cd**), as well as the percentage of explained variance (PEV), are also indicated. LOD values higher than genome-wide significant LOD scores are indicated in bold.

Trait	LG	cM	Marker	LOD	ac	ad	bc	bd	PEV
(CaR-CaL/CaR)_S	4c (R)	9.96	CR23,750	3.31	−1.48	−1.44	−2.30	−2.25	30.40
(ClR-ClL/ClR)_S	4c (R)	20.96	CR23-CR28	**4.21**	0.18	0.55	−0.49	−0.57	37.00
C_L_S	4c (R)	19.96	CR23-CR28	2.50	39.62	40.55	38.86	38.37	24.00
Ca_L_C	3a	0.00	CL2.26,395	2.49	25,124.90	22,355.50	20,850.10	21,678.50	23.90
Ca_L_S	4c (R)	10.96	CR23,750	**4.50**	18,996.90	18,213.40	25,142.40	23,567.90	39.00
Cl_L_S	4c (R)	19.96	CR23-CR28	**4.58**	57.72	36.56	121.88	116.64	39.40
Cl_R_S	7	128.19	CR76,1400	3.86	77.15	79.01	79.29	94.48	34.50
dCa_L	4c (R)	7.00	15R-CR23	2.95	−0.10	−0.10	0.12	0.03	27.70
dCl_L	4c (R)	11.27	CR23-CR28	**5.82**	6.63	4.61	13.60	15.66	47.20
dCl_R	7	128.19	CR76,1400	3.86	76.15	78.01	78.29	93.48	34.50
Fe_L_S	2 (Pr)	239.94	Mybg2,210	3.94	527.66	161.57	267.11	137.05	35.10
Fe_L_S	4c	0.00	15R,750	2.36	223.31	189.91	486.04	215.06	22.80
Fe_R_C	7	60.62	CL1.35-COR15	**4.58**	1147.92	750.15	942.96	3678.05	39.50
Fe_R_C	7	83.01	24R,950	3.68	1135.87	968.40	972.75	2557.25	33.20
Fe_R_S	3a	7.00	CL2.26,395	2.53	411.11	977.91	963.37	802.94	24.20
Fe_R_S	3b (R)	6.98	CR71,310	2.78	1004.75	778.82	756.57	461.35	26.30
FRDW_C	2	153.08	CR19,370	3.47	5.70	4.12	8.26	6.63	31.70
FRDW_C	4c (R)	7.00	15R-CR23	2.88	4.11	5.29	6.73	7.24	27.10
K_L_C	7 (R)	0.00	Myc2(HaeIII),480	3.67	49,804.20	56,643.20	43,620.80	46,139.00	33.10
K_L_C	10+5b	46.81	TAA41,160	**4.31**	53,039.40	41,858.30	44,812.40	52,128.20	37.70
K_L_C	3a (Pr)	34.57	CR31,100	2.98	38,263.80	54,458.60	49,637.80	47,656.50	27.90
K_L_S	7	24.43	CR41-CR20	3.40	47,939.30	41,816.10	38,534.00	44,202.00	31.10
K_L_S	4b (R)	66.24	CR3,320	3.57	47,744.90	42,798.50	40,029.70	37,750.10	32.40
N_R_C	10+5b	35.45	5F6R,1550	3.57	2.36	2.12	2.08	2.06	32.40
N_R_S	2	153.08	CR19,370	2.80	1.80	1.77	1.94	1.87	26.40
Na_R_C	7	28.43	CR20	3.82	1218.08	655.55	720.82	954.38	34.20
Na_R_S	7 (R)	83.01	24R,950	3.93	3046.92	3083.47	3743.36	4692.49	35.00
TRDW_C	2	153.08	CR19,370	3.02	18.82	16.53	23.37	20.40	28.20
TRDW_C	4c (R)	23.96	CR23-CR28	3.45	14.06	20.32	22.04	20.98	31.50

**Table 4 ijms-24-15759-t004:** List of candidate genes (mRNA) that might be involved in leaf Cl^−^ exclusion/accumulation at QTL *LCl-6* downloaded from the *C. clementina* genome database at https://phytozome.jgi.doe.gov accessed on 15 January 2022 between markers 15R,750 and CR28,270. The starting physical position in bp, a description, and comparison(s), where differential transcripts (differentially expressed genes, DEG) have been detected, are indicated (Appendix A). Comparisons 107_0 vs. 90_0, 107_15 vs. 90_15, and (107_15 + 107_0) vs. (90_15 + 90_0) are coded as 1, 2, and 3, respectively.

mRNA	Start	DEG in	Description
Ciclev10013718m	9543190		PTHR23515:SF3—HIGH AFFINITY NITRATE TRANSPORTER 2.5
Ciclev10011188m	10089075		PTHR10217//PTHR10217:SF494—VOLTAGE AND LIGAND-GATED POTASSIUM CHANNEL
Ciclev10012379m	10374766		PTHR19139:SF167—AQUAPORIN PIP2-1-RELATED
Ciclev10012633m	10396990		PTHR19139:SF167—AQUAPORIN PIP2-1-RELATED
Ciclev10012375m	10419232	3	PTHR19139:SF167—AQUAPORIN PIP2-1-RELATED
Ciclev10011234m	12362572		PTHR19241:SF258—ABC TRANSPORTER G FAMILY MEMBER 17-RELATED
Ciclev10013485m	12469611	3	PTHR19241:SF258—ABC TRANSPORTER G FAMILY MEMBER 17-RELATED
Ciclev10011167m	12478048	3	PTHR19241:SF258—ABC TRANSPORTER G FAMILY MEMBER 17-RELATED
Ciclev10011147m	12497442	3	PTHR19241:SF258—ABC TRANSPORTER G FAMILY MEMBER 17-RELATED
Ciclev10010954m	12508551	3	PTHR24093:SF289—CALCIUM-TRANSPORTING ATPASE 1
Ciclev10011060m	13305253	3	PTHR32468:SF10—CATION/H(+) ANTIPORTER 20
Ciclev10011092m	13371258		PTHR32468:SF10—CATION/H(+) ANTIPORTER 20
Ciclev10011096m	13412581		PTHR32468:SF10—CATION/H(+) ANTIPORTER 20
Ciclev10011745m	13446344	1, 2, 3	PTHR19444—UNC-93 RELATED—Major facilitator superfamily protein
Ciclev10013496m	15046032		PTHR11654:SF181—PROTEIN NRT1/ PTR FAMILY 2.8 (NPF2.3)
Ciclev10011341m	15116860	1, 3	PTHR11654:SF79—PROTEIN NRT1/ PTR FAMILY 5.5-RELATED (NPF5.12)
Ciclev10011514m	15540132		PTHR11662:SF235—ANION TRANSPORTER 3, CHLOROPLASTIC-RELATED
Ciclev10011381m	15747188	1, 3	PTHR11654//PTHR11654:SF125—OLIGOPEPTIDE TRANSPORTER-RELATED (NPF8.1)
Ciclev10013821m	15754314	3	PTHR11654//PTHR11654:SF125—OLIGOPEPTIDE TRANSPORTER-RELATED (NPF8.2)
Ciclev10013337m	15843855	1, 2, 3	PTHR11654:SF79—PROTEIN NRT1/ PTR FAMILY 5.5-RELATED (NPF5.9)
Ciclev10013636m	15862182		KOG1237—H+/oligopeptide symporter (NPF5.8)
Ciclev10013488m	15867971		PTHR11654:SF79—PROTEIN NRT1/ PTR FAMILY 5.5-RELATED (NPF5.10)
Ciclev10012384m	15896416		PTHR19139:SF169—AQUAPORIN PIP1-4-RELATED

**Table 5 ijms-24-15759-t005:** Number of differentially expressed genes (DEGs) per comparison, depending on the method used for their assessment, iDEP versus RSeqFlow, using the eBayes or Treat procedure, for the whole root transcriptome or for genes in the *LCl-6* QTL only. The *C. clementina* genome was used as reference for RNAseq readings here, and the Treat procedure was chosen for further analyses of DEGs.

Comparison	DEGs iDEP	DEGs RSeqFlow Transcriptome		DEG RSeqFlow QTL	
	Transcriptome	eBayes	Treat	eBayes	Treat
107_15 vs. 107_0	7	0	0	2	0
90_15 vs. 90_0	45	0	0	0	0
107_0 vs. 90_0	1201	1955	344	62	21
107_15 vs. 90_15	670	881	297	16	8
(107_15 + 107_0) vs. (90_15 + 90_0)	1501	5505	2400	130	92
(107_15 + 90_15) vs. (107_0 + 90_0)	42	0	0	14	0
(107_15 − 90_15) vs. (107_0 − 90_0)	na	0	0	0	0

## Data Availability

The RNA sequences will be available on the Bioproject (PRJEB61142) platform.

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
