# Peer review of "Combining Genetic and Transcriptomic Approaches to Identify Transporter-Coding Genes as Likely Responsible for a Repeatable Salt Tolerance QTL in Citrus"

_ijms, 2023, doi:10.3390/ijms242115759_

Round 1
Reviewer 1 Report
Comments and Suggestions for Authors I have the following comments need to be addressed: (1) Did the author replicate the phenotypes in different locations or years? (2) Also, do the selected markers work for other citrus cultivars, and (3) do they correlate to salt tolerance in other citrus cultivars or only in the mapping population? Comments on the Quality of English LanguageThere are several errors in this manuscript and so need to be revised carefully.
Author Response
Answers to Reviewer 1:
(1. Did the author replicate the phenotypes in different locations or years? ) Yes, two independent experiments were run at two locations (Valencia and Malaga), two years (2013-14 and 2019-20, respectively). Despite being very different experiments from each other (the former, GP, consisted of nucellar seedlings of the mapping population as rootstocks of a satsuma mandarin, and the latter, just nucellar seedlings without grafting, NG), phenotypes at common traits between experiments showed generally the same trend. To note this, and following a suggestion by reviewer 2, we have included a new table (Table 2) and a comment on this for the main trait leave Cl- (lines 159-162).
(2. Also, do the selected markers work for other citrus cultivars) As we have shown in this study, selected markers work fine for Citrus reshni and Poncirus trifoliata cultivars “Flying Dragon” and “Rich” which were different for NPF5.9 (Ptrif.0006s0823). We are now genotyping accessions from the IVIA citrus germplasm Bank to set up a core collection. Up to now, markers have worked fine in tested species.
(3. do they correlate to salt tolerance in other citrus cultivars or only in the mapping population?) This study shows that some genotypes at selected markers are significantly associated with salt tolerance (leaf Cl- exclusion) in both experiments and in a previously reported one using the same mapping population. Regarding other citrus cultivars, it has to be tested. For this purpose, we have set up a core collection where salt tolerance is studied more efficiently, similarly as we did regarding Citrus Tristeza Virus resistance (Bernet et al. 2008, https://doi.org/10.1111/j.1439-0523.2008.01506.x).
(4. There are several errors in this manuscript and so need to be revised carefully) We have tried to correct all errors in the manuscript. Changes in the manuscript are in blue.
Reviewer 2 Report
Comments and Suggestions for Authors
1. Improve the introduction section with recent studies, include studies related to the same crop or related species, and write the major scope of this study.
2. Related to phenotype data; Summarize one main table, it is not okay to present all the phenotype data as supplementary materials
3. Both results and discussion sections look like the dissertation thesis, I suggest summarizing them. And present in short and brief.
4. The same issue in methodology also. For instance, in RNA-sequencing and bioinformatics analyses, kindly divide into subheadings and describe the methodology.
Plant materials and RNA isolation
Library preparation and sequencing
Data processing
GO and KEGG analyses
5. If you validate the candidate genes by qRT-PCR analysis, the results will be more reliable.
6. Overall writing and presentation of the manuscript need subsequent improvement
Comments on the Quality of English LanguageModerate editing of English language required
Author Response
Answers to reviewer 2:
- We have improved the introduction, as suggested, by including a missing, relevant reference (Zhao et al. 2022) in lines 87-90, and the major scope of the study (lines 107-113).
- We have now included a table showing the means for the parents, the reference salt-sensitive and tolerant hybrids, and the whole population, in both experiments (GP and NG), under both conditions (Control and Salinity). Thus, the stability of phenotypes can be checked, as it is mentioned in lines 159-162, following a comment by reviewer 1.
- This manuscript describes a very complex study in citrus rootstocks based on two salt tolerance experiments where several physiological and agronomic traits were genetically analyzed to focus attention on a set of transporter-coding genes from a genome-wide transcriptomic analysis. Since results have to be presented clearly in the form of tables and graphics (mostly as supplementary materials) for a scientific community as broad as possible, we have not been able to reduce this section. In the discussion section, we have just noted relevant findings and commented on related results previously reported in citrus when available. We have now presented conclusions in brief in a separate section (lines 778-795).
- Subheadings have been included as suggested. Changes in the manuscript are in blue.
- The added value of a qPCR when using 3 biological replicates per sample is very low (Coenye, 2021). In fact, some bioinformaticians, such as David L. Adelson (https://orcid.org/0000-0003-2404-5636), claim that the use of 3 or more biological replicates in RNAseq generally provides more accurate assessments than q-PCR (https://www.researchgate.net/post/Why_is_qPCR_required_to_validate_RNAseq_result). We have further justified this issue in the text (lines 483-487).
- We have reviewed the manuscript following comments by reviewers to improve it. We appreciate their suggestions.
- English proofreading of the manuscript was carried out by a specialist, Michael O’Shea (line 925).
Round 2
Reviewer 2 Report
Comments and Suggestions for Authors
Congratulations to the authors. I recommend the manuscript for publication to IJMS.